# Tolerance of Human Fibroblasts to Benfo-Oxythiamine In Vitro

**DOI:** 10.3390/ijerph19074112

**Published:** 2022-03-30

**Authors:** Ming Yan, Ralf Smeets, Martin Gosau, Tobias Vollkommer, Sandra Fuest, Eva Stetzer, Lan Kluwe, Johannes F. Coy, Simon Burg

**Affiliations:** 1Department of Oral and Maxillofacial Surgery, University Medical Center Hamburg-Eppendorf, 20246 Hamburg, Germany; cnming.yan@hotmail.com (M.Y.); r.smeets@uke.de (R.S.); m.gosau@uke.de (M.G.); t.vollkommer@uke.de (T.V.); kluwe@uke.de (L.K.); 2Department of Oral and Maxillofacial Surgery, Guiyang Hospital of Stomatology, Guiyang 050017, China; 3Department of Oral and Maxillofacial Surgery, Division of “Regenerative Orofacial Medicine”, University Medical Center Hamburg-Eppendorf, 20246 Hamburg, Germany; s.fuest@uke.de; 4Benfovir AG, 64293 Darmstadt, Germany; stetzer@benfovir.com (E.S.); coy@benfovir.com (J.F.C.)

**Keywords:** benfo-oxythiamine, fibroblast, SARS-CoV-2, virus replication

## Abstract

Objectives: To explore the potential application of B-OT in the aspiration tract. Materials and Methods: We conceived and optimized an in vitro model simulating the mouth-washing process to assess tolerance to B-OT on primary human gingival fibroblasts. Cells derived from 4 unrelated donors were flushed with medium containing drugs of various concentration for one minute twice daily for 3 days. Results: No effect was seen on the cells up to 1000 µM B-OT. In addition, we treated the cells with B-OT permanently in medium, corresponding to a systemic treatment. No effect was seen by 10 µM B-OT and only a slight reduction (approximately 10%) was seen by 100 µM B-OT. Conclusions: Our results suggest good tolerance of oral cells for B-OT, favoring the further development of this antiviral reagent as a mouth-washing solution and nasal spray.

## 1. Introduction

Effective antiviral drugs against SARS-CoV-2 without severe side effects are urgently needed. A recent study reported that the pentose phosphate pathway is remarkably deregulated upon SARS-CoV-2 infection. Fructose-6P, intermediates of this pathway, can be converted by transketolase via a non-oxidative bypass to ribose-5-phosphate, which is required for the synthesis of nucleic acids [1,2,3]. In concordance, transketolase was indeed increased in SARS-CoV-2 infected cells. In line with the SARS-CoV-2-dependent upregulation of the pentose phosphate pathway (PPP) and its key enzyme transketolase, a dose-dependent inhibition of SARS-CoV-2 replication was detectable when non-cytotoxic concentrations of the transketolase inhibitor benfo-oxythiamine (B-OT) were applied (FFM1 (IC_50_: 0.21 mM) and FFM7 (IC_50_ 0.76 mM)) [2].

Due to the lack of effective drugs to treat COVID-19 patients, B-OT has been used in a compassionate-use setting to treat a hospitalized COVID-19 patient with confirmed SARS-CoV-2-related pneumonia in both lungs and oxygen deprivation. Prior to B-OT treatment, severe damage to the lungs with marked infiltrates of viral pneumonia was observed. Oral administration of a daily dose of 6 mg for 7 days completely suppressed SARS-CoV-2 replication (PCR negative), showed a marked decrease in the previously pronounced infiltrates, and resulted in the rapid clinical recovery of the patient within 7 days. In a 70 kg patient, 6 mg B-OT corresponds to a serum peak concentration of 500 pM.

In addition, the upper respiratory tract, starting with the nasal and oral cavities, is a major route for the entry of pathogens into the body [4]. SARS-CoV-2 infects epithelial cells of the upper respiratory tract (URT; including the nasal, oral and throat) and the lungs (bronchi and lung alveoli) [5]. That is why all tests involve oro/nasopharyngeal swabbing for SARS-CoV-2. It is intuitively obvious that the oral cavity, along with the nasal cavity, being proximal to the airway, would be the most direct route for microbes to enter the airway. As SARS-CoV-2 infection occurs via the aspiration track, anti-viral drugs administered as a nasal spray and mouth-washing solution may provide an effective and easy local measure for prevention and treatment for the initial stages [6]. The upper respiratory tract is normally considered to include the nasal cavity, the oral cavity, the oro- and nasopharynx, and the larynx [7]. The oral and nasal cavities are covered by the same mucosal epithelium that starts at the beginning of the aero-digestive tract [8,9]. Therefore, these cells should be interpreted as the same cells. In addition, the upper respiratory tract, starting with the nasal and oral cavities, is a major route for the entry of pathogens into the body [10]. 

The aim of this study is to establish an in vitro model to evaluate the tolerance of human fibroblasts to benfo-oxythiamine. In the present study, we cultured human primary gingival cells by flushing and shaking for 1 min with a B-OT-containing medium twice daily to explore the potential toxic effect of this treatment on the cells. 

## 2. Materials and Methods

### 2.1. Primary Cells Were Obtained

Human primary gingival fibroblasts were cultured from tissues attached to teeth that were extracted for medical or dental indication [11]. Recovering these biowastes for use as a specimen in cell culture in an anonymized manner is not under ethical regulation. However, the protocol was registered to the local privacy protection authority and all patients gave their written consent. The soft tissues on the teeth were scratched off and further cut into small pieces which were placed as explants onto the surface of culture dishes. After several days, the cells grow and migrate from the explants. Cells from four unrelated teeth in their 3–5 passages were used for the experiments. 

### 2.2. Cell Culture

For the treatment, 2000 cells were seeded into one well of a 96-plate. For each drug and concentration, five replicates were set up: four for the viability assay and one for the nuclei staining using DAPI. The cells were cultured for one day and the flushing treatment (after the drug was added to the culture plates, it was shaken for one minute) was started on the 3rd day. The flushing was carried out twice a day by removing the medium to wells of a new plate and adding 20 µL of medium with B-OT (Tavargenix GmbH, Darmstadt, Germany; benfovir AG, Darmstadt, Germany), preincubated B-OT overnight in serum-containing medium or oxythiamine at various concentrations. B-OT is a prodrug of oxythiamine and is inverted into oxythiamine in the presence of serum. Ketoconazole at 300 µM was used as a toxic control. After 1 min, the drug-containing medium was removed and discarded. The previous medium was pipetted back to the corresponding wells of the plate in treatment. 

In contrast, reducing the starting cell density led to a loss of tolerance of the cells against the manipulation of flushing twice daily. The experimental setting was therefore optimized to 2000 cells/well at seeding, an overnight growth of the cells, and a treatment period of 3 days (Figure 1). 

### 2.3. Cell Proliferation Assay

The flushing was continued for 3 days and on the next day of the last flushing, viabilities of the cells were measured using the XTT assay kit (Roche Diagnostics GmbH, Mannheim, Germany). From the 4 replicates flushed with medium without any drug, the mean extinct at 450 which reflects the viability of the cells was calculated and used to normalize all other extinct values. Using the normalized values, the mean and standard deviation of the viability of the cells flushed with each drug at each concentration were calculated. 

For the cells of the 4th donor, we also included a treatment with B-OT permanently in the medium parallel to the wash treatment, both for 1, 2 and 3 days.

### 2.4. Statistical Analysis

Statistical significance was analyzed by a *t*-test (SPSS 17.0 software, Chicago, IL, USA). The data were presented as the mean ± standard deviation (mean ± SD). All tests with a *p*-value of less than 0.05 were considered statistically significant.

## 3. Results

Flushing with the medium containing up to 1 mM B-OT or oxythiamine twice daily did not affect the viability of the cells for all three cultures, which were all compatible with that of the cells flushed with the medium of no drugs. The next concentration of 10 mM reduced the viability to approximately 50% (*p* < 0.05) compared to the no-drug controls (Figure 2). Ketoconazole at 300 µM also reduced the cell viability to approximately 50% (*p* < 0.05), indicating that this in vitro model is suitable to detect toxic effects. There was no statistical difference between the B-OT with and without pre-incubation in serum-medium (*p* > 0.05). A similar or slightly better tolerance was also observed for oxythiamine. 

Moreover, when B-OT was permanently kept in the medium for 3 days, no effect on cell viability was seen up to 10 µM (*p* > 0.05). For 100 µM, only a slight reduction (*p* < 0.05) (approximately 10%) was seen (Figure 3).

## 4. Discussion

In the present study, we analyzed the tolerance of human fibroblasts to Benfo-Oxythiamine in vitro using an antiviral solution. We used primary human gingival fibroblasts derived from fresh tissues of extracted teeth. Compared to established cell lines which are standard in most drug testing studies, primary cells in culture better reflect the physiological environment in the human body. In addition, our cells were derived from the oral cavity—the area the intended mouth-washing solution will come into direct contact with. We optimized the experimental setting including: (1) a starting density of 2000 cells/well, (2) one-day growth before treatment and (3) a treatment period of 3 days. The setting is adopted to fast-growing cells which is the case in the present study. For other types of cells, the parameters may need to be adjusted. 

Establishing this in vitro model allowed us to evaluate the tolerability of B-OT use on primary human gingival fibroblasts. B-OT represents a novel prodrug releasing OT, a well-known inhibitory thiamine derivative and transketolase inhibitor. 

The use of a transketolase inhibitor was previously considered by experts to be an unfeasible therapeutic approach because too many or too strong side effects for the host cell were expected. As essential enzymes of the PPP, transketolases are involved in the production of R5P in the cell, which is required for mRNA formation as well as for DNA repair and DNA synthesis during cell division. A sufficient, basic supply of R5P to the cell, therefore, appears indispensable to ensure cell viability. The TKT transketolase represents a housekeeping gene; even the deletion of one of the two TKT genes in the genome leads to severe problems. 

However, in addition to the TKT transketolase, a transketolase-like gene (transketolase-like 1; TKTL1) was discovered as early as 1996 [12], although its possible function was unknown for a long time. It was only in 2019 that Li et al. discovered that TKTL1 is the transketolase that controls the production of high levels of R5P during the cell cycle prior to cell division [13]. While TKT-TKT homodimers ensure the maintenance of the basic R5P supply, an upregulation of TKTL1 and the formation of TKTL1-TKT heterodimers occur prior to cell division. This alters the enzymatic activity leading to a strong increase in R5P synthesis. These findings by Li et al. change the old paradigm of cell cycle control and the production of R5P as a DNA building block. Accordingly, R5P synthesis in cells is controlled differently depending on the stage of the cell cycle: while the housekeeping function of the cell requires R5P synthesis at comparatively low levels (TKT-TKT homodimer mediated), genome duplication prior to cell division requires a transient but strong increase in R5P synthesis (TKT-TKTL1 heterodimer mediated). Furthermore, high R5P levels in the cell are also needed for DNA repair mechanisms especially in the case of high DNA damage due to DNA damaging chemotherapies or radiotherapies (gamma, alpha, and beta radiation). 

Viruses also require high levels of R5P for replication and multiplication, exploiting the host’s metabolism. One of the most powerful ways to increase host cell R5P production is through the activation of TKTL1, which leads to increased R5P synthesis via TKT-TKTL1 heterodimer formation, thus facilitating the production of new viral RNA. For SARS-CoV-2, it has already been shown that PPP and TKT are upregulated in SARS-CoV-2-infected cells [2,3], indicating that the virus manipulates this metabolic pathway.

Furthermore, Codo et al. observed an altered metabolism in SARS-CoV-2-infected monocytes: SARS-CoV-2 infection induced the stabilization of hypoxia-inducible factor-1 alpha (HIF1α) and promoted sustained aerobic glycolysis, i.e., the metabolism of glucose via glycolysis rather than oxidative phosphorylation despite the presence of sufficient oxygen [1].

This metabolic switch to oxygen-independent glycolysis (so-called aerobic glycolysis—glycolysis despite the presence of oxygen = Warburg effect) is associated with facilitated viral replication and promotes cytokine production and subsequent T-cell dysfunction and lung epithelial cell death [1]. As a metabolic switch from oxidative metabolism (OxPhos) using mitochondria to aerobic glycolysis is accompanied by an increased glucose demand of the cells, this might also explain the fact that high blood glucose levels (diabetes patients and non-diabetes patients) are associated with an increased risk of severe COVID-19 and increased glucose levels in the sera of COVID-19 patients that are correlated with a poor prognosis [14,15,16]. A meta-analysis by Chen et al. provides evidence that severe COVID-19 infection is associated with elevated blood glucose levels, and the authors emphasize the need for effective monitoring of blood glucose to improve prognosis in patients infected with COVID-19 [15].

Interestingly, this modulation of cellular metabolism towards aerobic glycolysis has already been shown for malignant cancer cells [17,18]. Overexpression of TKTL1 has been associated with poor prognosis, invasivity, and metastasis and has been described in numerous cancer types including breast, lung, colonic, urothelial, esophageal, gastric, laryngeal cancer, melanoma, and oral squamous cell carcinomas (OSCC) and contributes to the development of a malignant phenotype via the stabilization and accumulation of HIF1α and increased aerobic glycolysis [18,19,20,21,22,23,24,25,26,27,28]. The causal role that TKTL1 plays in cell cycle progression and R5P delivery was demonstrated using siRNA experiments to inhibit TKTL1, which resulted in cell cycle arrest [29]. In addition, siRNA inhibition of TKTL1 also overcame preexisting resistance to cisplatin, paclitaxel, and radiotherapy [30,31,32,33].

Consistent with these experiments, OT, the active component of B-OT, has already been shown to reduce cancer cell proliferation and tumor growth in both cell culture experiments and xenograft models of various cancers, both as a monotherapy or in combination with chemotherapies [33,34,35,36,37,38,39]. 

Taken together, these data demonstrate that TKTL1 transketolase plays an important role in the regulation of the cell cycle and R5P accumulation and contributes to the stabilization of the HIF-1alpha/aerobic glycolysis axis. These mechanisms appear to promote cancer cell malignancy as well as viral replication and cytokine expression in monocytes following SARS-CoV-2 infection.

Substances that inhibit the activity of transketolases, such as the thiamine antagonist B-OT and its active compound OT, are likely to prevent the accumulation of R5P, the essential building block for viral replication, in the cell. Although OT is unlikely to selectively inhibit the TKTL1-TKT heterodimer, it is likely to result in a marked reduction in R5P synthesis. Published data for OT, as well as unpublished preclinical data for B-OT/OT, show that B-OT/OT can be used in such a way that R5P formation can be suppressed to an extent that both RNA and DNA production and, accordingly, cell division and virus replication can be inhibited without causing irreversible damage to the cell. Irreversible damage is the result of too high doses or too long a treatment period.

In our study, we designed and optimized an in vitro model simulating the mouth-rinsing process to explore a possible application of B-OT in the aspiration tract and to investigate the tolerance of primary human gingival fibroblasts to B-OT. Cells derived from four unrelated donors were flushed with medium containing drugs of various concentration for one minute twice daily for 3 days. No effect was seen on the cells up to 1000 µM B-OT. In addition, we treated the cells with B-OT permanently in medium, corresponding to a systemic treatment. No effect was seen by 10 µM B-OT and only a slight reduction (approximately 10%) was seen by 100 µM B-OT. 

As described in the introduction, a recent compassionate clinical use of B-OT for a hospitalized COVID-19 patient demonstrated the high potency of B-OT with an efficacious dose of 6 mg daily, corresponding to a peak serum concentration of approximately 500 pM. This effective dose range of BOT is 10^6^–fold below the 1000 µM which did not show any toxic effect on cultured primary gingival cells in the present study. Even when B-OT was kept in medium permanently during the 3-day treatment period, the cells were completely tolerant to 10 µM and only slightly affected by 100 µM.

## 5. Conclusions

In conclusion, our results demonstrate excellent tolerance of human primary gingival cells to B-OT flushing treatment at up to 1 mM, suggesting good tolerance of oral cells for B-OT, favoring the further development of this antiviral reagent as a mouth-washing solution and nasal spray for antiviral treatment. In addition, a local, oral application for the treatment of oral tumors could also be conceivable.

## Figures and Tables

**Figure 1 ijerph-19-04112-f001:**
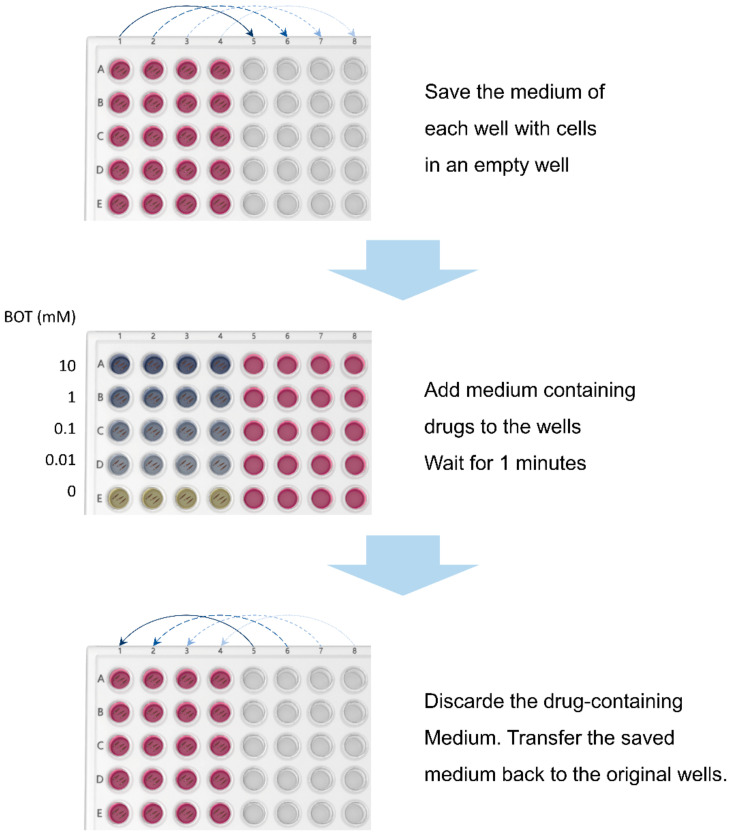
An illustration of the procedure of the in vitro model for mouth washing. Cells were seeded into wells of a 96-plate plate. For each washing, the medium of each well containing cells was saved in an empty well; medium containing various drugs of various concentrations was added to the corresponding wells. After 1 min, the drug-containing medium was discarded and the saved medium was transferred back to the original wells. The treatment was carried out twice daily.

**Figure 2 ijerph-19-04112-f002:**
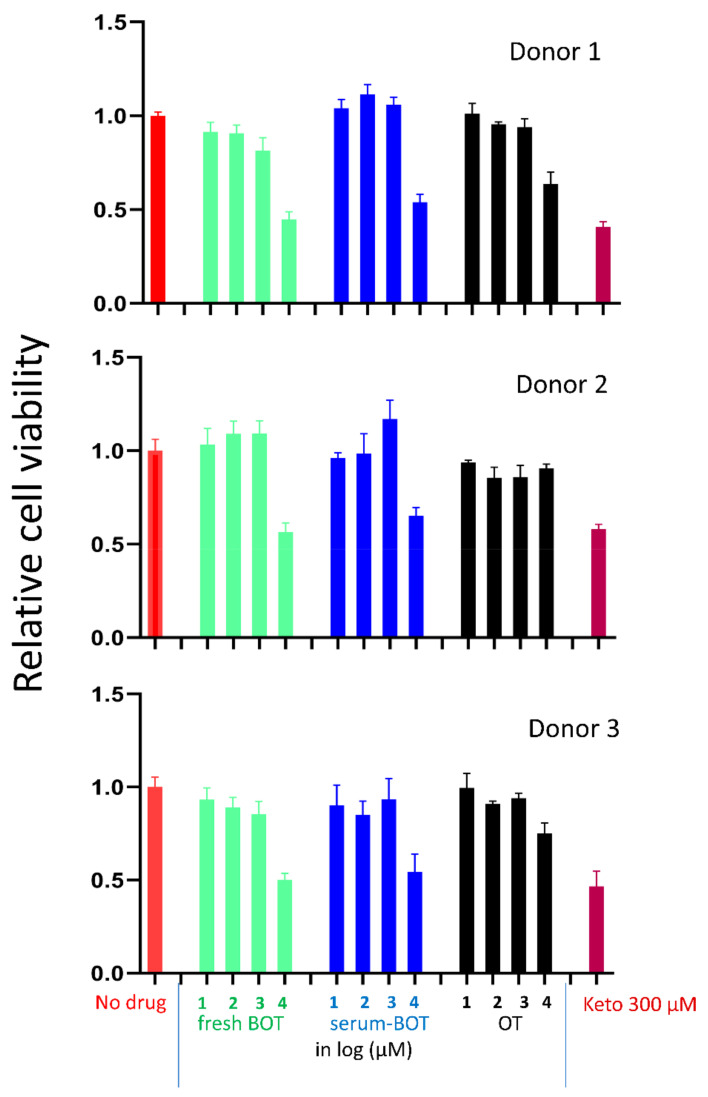
Relative viability of cells in 3 cultures of primary human gingival fibroblasts derived from 3 unrelated donor teeth. Fresh B-OT: fresh benfooxythamine; serum B-OT: benfooxythamine preincubated in serum-containing medium overnight at room temperature. OT: oxythiamine. Keto: Ketoconazole.

**Figure 3 ijerph-19-04112-f003:**
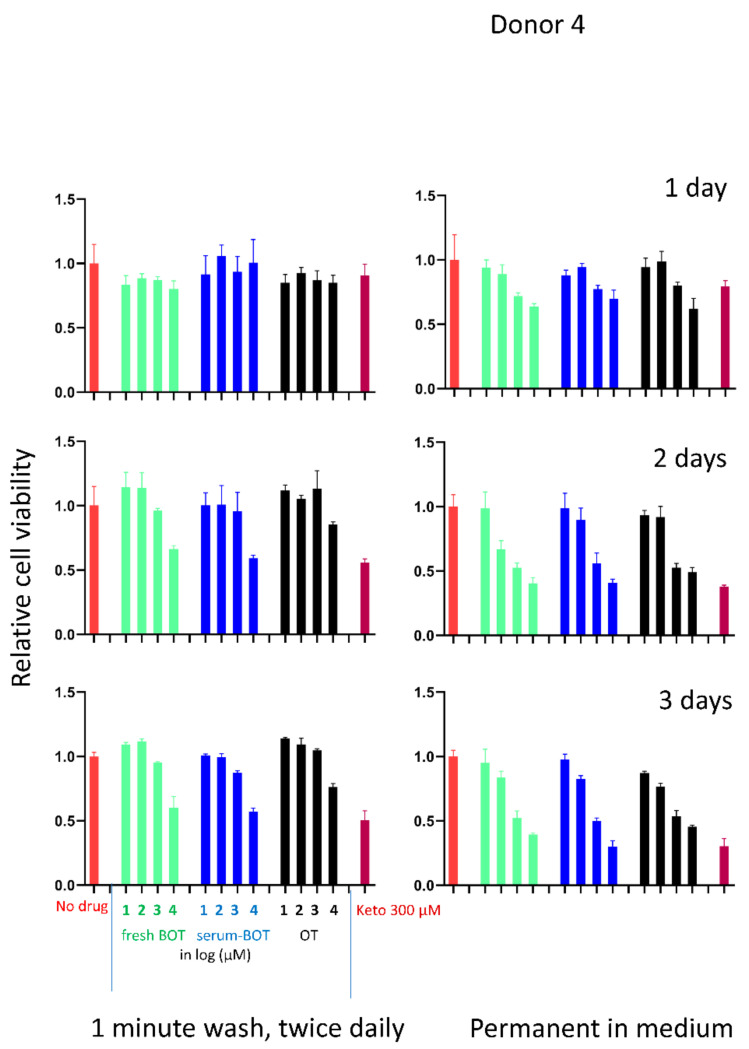
Conventional treatment with B-OT in the medium over the treatment period of 3 days. The gingival cells were derived from a 4th donor tooth. The flushing treatment was also carried out in parallel for comparison. In addition, we measured the cell viabilities also for treatment of 1 and 2 days in both the flushing and conventional mode.

## Data Availability

Data available on request due to restrictions eg privacy or ethical.

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
