# Peer review of "Tolerance of Human Fibroblasts to Benfo-Oxythiamine In Vitro"

_ijerph, 2022, doi:10.3390/ijerph19074112_

Round 1
Reviewer 1 Report
Thank you for taking my previous remarks into consideration.
A few points of detail to be taken up please.
Line 37-45: What are the references?
In addition, the upper respiratory tract, starting with the nasal and oral cavities, is a major route for entry of pathogens into the body[4]
Reference 4 is incorrect: Saferali, A.; Tang, A.C.; Strug, L.J.; Quon, B.S.; Zlosnik, J.; Sandford, A.J.; Turvey, S.E. Immunomodulatory function of the cystic fibrosis modifier gene BPIFA1. PLoS One 2020, 15, e0227067, doi:10.1371/journal.pone.0227067.
I suggest: doi: 10.3389/fmicb.2021.786042
Line 52-59: Please include references
All tests with a p-value of less than 0.05 were considered statistically significant. Unless I am mistaken I do not see the results of the tests on the figures (*) or in the text
Line 96-114: It is from the Method section and not Results
What are the limitations of your study?
Line 120: There was no obvious difference. What does obvious mean? Statistical?
Line 244: In conclusion, our results "demonstrate". This type of study and analysis does not demonstrate. "Underline perhaps?"
Author Response
Thank you for giving your suggestion again, you are indeed an expert in this field
Line 37-45: What are the references?
Reply:This part is not yet published, and it is a medical record that we use for clinical treatment. The manuscript is now under preparation, so we cannot use the cited literature.
In addition, the upper respiratory tract, starting with the nasal and oral cavities, is a major route for entry of pathogens into the body[4]
Reference 4 is incorrect: Saferali, A.; Tang, A.C.; Strug, L.J.; Quon, B.S.; Zlosnik, J.; Sandford, A.J.; Turvey, S.E. Immunomodulatory function of the cystic fibrosis modifier gene BPIFA1. PLoS One 2020, 15, e0227067, doi:10.1371/journal.pone.0227067.
I suggest: doi: 10.3389/fmicb.2021.786042
Reply:done as suggestion
Line 52-59: Please include references
Reply:done as suggestion
All tests with a p-value of less than 0.05 were considered statistically significant. Unless I am mistaken I do not see the results of the tests on the figures (*) or in the text
Reply: We added the p-value
Line 96-114: It is from the Method section and not Results
Reply :This section has been adjusted to the methods section
What are the limitations of your study?
Reply: We have placed the limitations section in the discussion. However, we recognize the limitations of our research. such as the absence of in vivo experiment. This study only explored the tolerance of human fibroblasts to Benfo-Oxythiamineas in an in vitro. The verification of our conclusion in vivo is still needed to be done in the future.
Line 120: There was no obvious difference. What does obvious mean? Statistical?
Reply:statistical significance. We have changed this sentence
Line 244: In conclusion, our results "demonstrate". This type of study and analysis does not demonstrate. "Underline perhaps?"
Reply:change the word "demonstrate" to "Underline"?
Reviewer 2 Report
Dear author,
Your article is interesting but loses scientific quality because you are trying to market your in vitro culture model as a mouthwash and focus on SARS-CoV-2.
The manuscript should be rewritten by removing the fact that your in vitro model mimics a mouthwash. Please also consider the following comments
Title
Please delete "mouth-washing model"
Introduction
- L35 Give the meaning of PPP
- L37 Give the meaning of FM1
- L55 Delete “mouth-washing”
- L56-59 This sentence should be modified because you have no proofs that your model corresponds to a mouth-washing process. Indeed, you only add the BOT and then change the medium. It could have been interesting to shake the plate maybe…
Material and methods
- The technique of culture of fibroblast has ever been published. Please add references
Results
- Figure 1: I am not sure of the interest of this figure. If the culture of fibroblasts has ever been published, delete this figure. If not, the characterization of the cells obtained by this technique of culture should be added.
- Figure 3: Why the x-axis is in log?
- L158 “seen(Fig. 4)..” Add onse space before “(“ and delete one point
Discussion
- “In the present study, we designed and optimized an in vitro model to simulate the mouth-washing process using an antiviral solution.” How have you proved that your model corresponds to the mouth-washing process?
If you have no evidences then reformulate this sentence. For example, “”In the present study, we analyzed the tolerance of human fibroblasts to Benfo-Oxythiamineas”
in vitro mouth-washing model
- L147 Delete “in a mouth-washing process” because you have no proof that your in vitro model corresponds to a mouth-washing process
- L228 “In our study, we designed and optimised an in vitro model simulating the mouth rinsing process to explore a possible application of B-OT in the aspiration tract and to investigate the tolerance of primary human gingival fibroblasts to B-OT.” You cultured human gingival fibroblast and not human fibroblasts of the aspiration tract. Do you have evidence of the similarity of cells?
- L236 Add the reference
Author Response
Your article is interesting but loses scientific quality because you are trying to market your in vitro culture model as a mouthwash and focus on SARS-CoV-2.
The manuscript should be rewritten by removing the fact that your in vitro model mimics a mouthwash. Please also consider the following comments
Title
Please delete "mouth-washing model"
Reply: done as suggestion
Introduction
- L35 Give the meaning of PPP Reply:pentose phosphate pathway (PPP)
- L37 Give the meaning of FM1 Reply: (FFM1 and FFM7) are Caco2 cells infected with two different SARS-CoV-2 isolates
- L55 Delete “mouth-washing” Reply: done as suggestion
- L56-59 This sentence should be modified because you have no proofs that your model corresponds to a mouth-washing process. Indeed, you only add the BOT and then change the medium. It could have been interesting to shake the plate maybe… Reply: done as suggestion
Material and methods
- The technique of culture of fibroblast has ever been published. Please add references
Reply: done as suggestion
Results
- Figure 1: I am not sure of the interest of this figure. If the culture of fibroblasts has ever been published, delete this figure. If not, the characterization of the cells obtained by this technique of culture should be added. Reply: we delete this part
- Figure 3: Why the x-axis is in log? Reply: Because in our experiments, the drug concentration units are mM,The four experimental groups, if converted to μM, would become 10 100 1000 10000. So these four experimental groups take log, which can be converted into numbers 1 2 3 4, so that the arrangement in x-axis is more concise.
- L158 “seen(Fig. 4)..” Add onse space before “(“ and delete one point . Reply: done as suggestion
Discussion
- “In the present study, we designed and optimized an in vitro model to simulate the mouth-washing process using an antiviral solution.” How have you proved that your model corresponds to the mouth-washing process?
If you have no evidences then reformulate this sentence. For example, “”In the present study, we analyzed the tolerance of human fibroblasts to Benfo-Oxythiamineas” in vitro mouth-washing model Reply: done as suggestion.
- L147 Delete “in a mouth-washing process” because you have no proof that your in vitro model corresponds to a mouth-washing process. Reply: done as suggestion.
L228 “In our study, we designed and optimised an in vitro model simulating the mouth rinsing process to explore a possible application of B-OT in the aspiration tract and to investigate the tolerance of primary human gingival fibroblasts to B-OT.” You cultured human gingival fibroblast and not human fibroblasts of the aspiration tract. Do you have evidence of the similarity of cells? Reply: Upper respiratory tract is normally considered to includethe nasal cavity, the oral cavity, the oro-and nasopharynx, and the larynx [1]. The oral and nasal cavities are covered by the same mucosal epithelium that starts at the beginning of the aero-digestive tract[2,3]. Therefore these cells should be interpreted as same cells.
In addition, the upper respiratory tract, starting with the nasal and oral cavities, is a major route for entry of pathogens into the body[4].
In contrast, a significant number of cells are easy to obtain because we are Oral and Maxillofacial Surgeon.
- L236 Add the reference Reply: This part is not yet published, and it is a medical record that we use for clinical treatment. The manuscript is now under preparation, so we cannot use the cited literature.
- George, S.C.; Hlastala, M.P. Airway gas exchange and exhaled biomarkers. Compr Physiol 2011, 1, 1837-1859, doi:10.1002/cphy.c090013.
- Kiyono, H.; Yuki, Y.; Nakahashi-Ouchida, R.; Fujihashi, K. Mucosal vaccines: wisdom from now and then. Int Immunol 2021, 33, 767-774, doi:10.1093/intimm/dxab056.
- Miao, M.; Peng, M.; Xing, Z.; Liu, D. Effect of Shuangjinlian mixture on oral ulcer model in rat. Saudi J Biol Sci 2019, 26, 790-794, doi:10.1016/j.sjbs.2019.02.005.
- Saferali, A.; Tang, A.C.; Strug, L.J.; Quon, B.S.; Zlosnik, J.; Sandford, A.J.; Turvey, S.E. Immunomodulatory function of the cystic fibrosis modifier gene BPIFA1. PLoS One 2020, 15, e0227067, doi:10.1371/journal.pone.0227067.

Reviewer 3 Report
- The narrative of "As SARS-CoV-2 infection occurs via the aspiration track, anti-viral drugs administered as nasal spray and mouth washing solution may provide an effective and easy local measure for prevention and treatment for the initial stages." can be reinforced. Explain why it works.
- 55-59 Please indicate specific goals.
- However, the protocol was registered to the local privacy authority protection and all pa-65 tients gave their written consent. Is there documentation of approval?
- Figure 1. BC, Ruler is very unclear.
- Figure 3. and Figure 4. Please include statistical analysis.
- The manuscript is very interesting and clinically valuable, which is great.
Author Response
- The narrative of "As SARS-CoV-2 infection occurs via the aspiration track, anti-viral drugs administered as nasal spray and mouth washing solution may provide an effective and easy local measure for prevention and treatment for the initial stages." can be reinforced. Explain why it works. Reply : As SARS-CoV-2 infection occurs via the aspiration track, anti-viral drugs administered as nasal spray and mouth washing solution may provide an effective and easy local meas-ure for prevention and treatment for the initial stages [1].Upper respiratory tract is nor-mally considered to includethe nasal cavity, the oral cavity, the oro-and nasopharynx, and the larynx [8]. The oral and nasal cavities are covered by the same mucosal epithelium that starts at the beginning of the aero-digestive tract[9,10]. Therefore these cells should be interpreted as same cells.
- 55-59 Please indicate specific goals. Reply: done as suggestion.
However, the protocol was registered to the local privacy authority protection and all pa-65 tients gave their written consent. Is there documentation of approval? Reply: Administrative permissions were acquired by our team to access the data used in our research. The study protocol was approved by the Hamburg University ethics committee that approved the study. No: REC 1712/5/2008. Accordingly, all teeth were coded with number and all personal identification of the patients were removed. All parent or guardian of participants provided written informed consent for using their teeth which otherwise would have been discarded as waste.
- Figure 1. BC, Ruler is very unclear. Reply: reviewer suggest us to delete figure 1. What should we do?
- Figure 3. and Figure 4. Please include statistical analysis. Reply: done as suggestion.
- The manuscript is very interesting and clinically valuable, which is great. Reply: Many thanks are due to you for helpful suggestion.
- Tzou, P.L.; Tao, K.; Nouhin, J.; Rhee, S.Y.; Hu, B.D.; Pai, S.; Parkin, N.; Shafer, R.W. Coronavirus Antiviral Research Database (CoV-RDB): An Online Database Designed to Facilitate Comparisons between Candidate Anti-Coronavirus Compounds. Viruses 2020, 12, doi:10.3390/v12091006.

Round 2
Reviewer 2 Report
Dear author,
Thank you for considering my comments.
Just one more concerning the title. Delete "in an"
Author Response
done as suggestion
This manuscript is a resubmission of an earlier submission. The following is a list of the peer review reports and author responses from that submission.
Round 1
Reviewer 1 Report
It is with interest, I read your manuscript.
The study was carried out to explore the potential application of B-OT in the aspiration tract. It was an in-vitro study whereby the mouth-washing process was simulated and tolerance of the developed B-OT on the gingival fibroblasts.
Right at the first glance, I have noticed minor English errors. For instance, in the abstract, line 15 the word 'we' should have been capitalized.
Then, in methods, line 113, the phrase 'After 1 minutes' should have been 'After 1 minute'.
My two major concerns are;
- I do not see sufficient evidence from your study to claim that it has a therapeutic effect on SARS-CoV-2. Also, your aim was only to check the tolerance of B-OT. It would be misleading if in your response you fail to show the claimed evidence (abstract; line 24).
- Why was the flushing started on the third day, and is there a specific reason or evidence to do the same?
- Even though some numerical results using calculations were provided, there was no statistical test mentioned in the methodology.
Author Response
We gratefully acknowledge for your helpful comments and suggestions.
Right at the first glance, I have noticed minor English errors. For instance, in the abstract, line 15 the word 'we' should have been capitalized.
Reply: done as suggestion.
Then, in methods, line 113, the phrase 'After 1 minutes' should have been 'After 1 minute'.
Reply: done as suggestion.
My two major concerns are;
I do not see sufficient evidence from your study to claim that it has a therapeutic effect on SARS-CoV-2. Also, your aim was only to check the tolerance of B-OT. It would be misleading if in your response you fail to show the claimed evidence (abstract; line 24).
Reply: This is a very good question you have asked. Before we started this study, one of our studies showed that B-OT has been used in a compassionate use setting to treat a hospitalised COVID-19 patient with confirmed SARS-CoV-2-related pneumonia in both lungs and oxygen deprivation. Prior to B-OT treatment, severe damage to the lungs with marked infiltrates of viral pneumonia was observed. Oral administration of a daily dose of 6mg for 7 days completely suppressed SARS-COV-2 replication (PCR negative), showed a marked decrease in the previously pronounced infiltrates, and resulted in rapid clinical recovery of the patient within 7 days (manuscript in preparation). (https://www.ots.at/presseaussendung/OTS_20211210_OTS0013/neuer-therapieansatz-gegen-sars-cov-2-erfolgreich-getestet)In a patient with 70kg body weight, 6mg B-OT corresponds to a serum peak concentration of 500pM.
Because this drug is effective against the SARS-CoV-2, we are conducting the current experiment to ensure that this drug does not harm normal human fibroblasts; in other words, the concentration of the drug that proves effective against the SARS-CoV-2 has no toxic effect on human cells. Of course, we can also delete this sentence.
Why was the flushing started on the third day, and is there a specific reason or evidence to do the same?
Reply: Because cells attachment usually needs 24 hours, the second 24 hours is a period of rapid and stable cell growth. Then cell were allowed to be cerried out drug treatment.
Even though some numerical results using calculations were provided, there was no statistical test mentioned in the methodology.
Reply: Based on your suggestions, we have added the following sections: Statistical significance was analyzed by t-test (SPSS 17. 0 software). The data were presented as mean ± standard deviation (mean±SD). All tests with a p-value of less than 0.05 were considered statistically significant.

Reviewer 2 Report
Before going further in my reviewing, I would like to have more information on 2 elements:
1 / How does your study relate to recent links (November-December 2021) that can be found on the web concerning research on Benfo-Oxythiamine and Covid-19 with a priori an English leadership and German university researchers?
How is it that you do not refer to this work which is at the pre-clinical stage (or it is the same research)?
2 / One of your authors is affiliated with Benfovir AG, which can also be found in the links mentioned above. (https://www.ots.at/presseaussendung/OTS_20211210_OTS0013/neuer-therapieansatz-gegen-sars-cov-2-erfolgreich-getestet). However, you indicate in the text that there is no conflict of interest.
Author Response
We gratefully acknowledge for your helpful comments and suggestions.
Before going further in my reviewing, I would like to have more information on 2 elements:
1 / How does your study relate to recent links (November-December 2021) that can be found on the web concerning research on Benfo-Oxythiamine and Covid-19 with a priori an English leadership and German university researchers?
Reply: Prof. Cinatl discovered the first SARS-CoV virus in 2003 and is a very eminent virologist and oncologist. He and his research group at the University of Frankfurt, in collaboration with the University of Kent, have shown that B-OT inhibits the replication of the SARS-CoV-2 virus in human cells. This study was published in October last year in the journal Metabolites, a MDPI journal.
Metabolites 2021, 11, 699. https://doi.org/10.3390/metabo11100699
The company benfovir provided him with the active substance B-OT for this research work.
How is it that you do not refer to this work which is at the pre-clinical stage (or it is the same research)?
Reply: After the publication of the study on the efficacy of B-OT against the SARS-CoV-2 virus, the preclinical development of B-OT was completed. E.g. it was shown that B-OT has no genotoxic effect and does not bind to receptors and nerve channels (e.g. hERG assay). B-OT has fulfilled all the requirements to now be evaluated in a clinical phase I with regard to its use as an orally administered, systemically acting inhibitory thiamine derivative. An application for a clinical phase I was submitted by benfovir AG in December and this was taken as an opportunity to publish the press release you mentioned.
The data regarding the exposure of epithelial cells to B-OT show that B-OT is suitable for use as a mouth-wash or nasal spray. As B-OT is a prodrug that releases the active ingredient oxythiamine (OT), previous studies on the antiviral effect of OT are of great importance. In these earlier studies, OT was shown to inhibit polio-, mumps-, influenza- and rhinoviruses. These studies, which were published a long time ago, and the key statements can be found in the attachment of my email. It could make sense to list and discuss these important historical studies on oxythiamine in the submitted manuscript.
These data, published in the last century, impressively support that the prodrug B-OT, as an OT-releasing agent, is potentially able to inhibit influenza and rhinoviruses as well.
The good tolerability of B-OT is the basis that this active ingredient can now be evaluated in clinical trials with regard to its use as a mouth-wash and a nasal spray.
2 / One of your authors is affiliated with Benfovir AG, which can also be found in the links mentioned above. (https://www.ots.at/presseaussendung/OTS_20211210_OTS0013/neuer-therapieansatz-gegen-sars-cov-2-erfolgreich-getestet). However, you indicate in the text that there is no conflict of interest.
Reply by Johannes F. Coy: As founder and shareholder of the pharmaceutical companies benfovir AG and TAVARGENIX GmbH, which preclinically developed the active substance B-OT, I naturally have a conflict of interest. I apologise that this was not stated in the submitted manuscript. In our new manuscript, we will correct this error

Reviewer 3 Report
Dear authors,
The conclusions of your article are not based on the results of this article. One major remark concerns the tittle which doesn’t reflect the study because you did neither analyzed cells infected with SARS-CoV-2 nor the use of mouthwash. Moreover, the number of samples is very low because you used only one sample for the results of the figure 4.
Tittle
- The title should be revised because “Benfo-Oxythiamine as potential mouth-wash and nasal spray 2 against SARS-CoV-2” is a hypothesis and not results of this study
Maybe one tittle such as “Tolerance of human fibroblasts to Benfo-Oxythiamineas in an in vitro mouth-washing model”
Abstract
- L14 Give the meaning of « B-OT”
- L16 “to assess tolerance” Please clarify the tolerance of what?
- L40 “cells [1,2].A human” Add one space after the point
- L54-90 This section should be summarized to be more clear
- The section clinical relevance that is not a mandatory of this journal should be deleted because it is purely hypothetic
Material and methods
- Add subheading
- The technique of culture of fibroblast has ever been published. Please add references
Results
- Figure 1: I am not sure of the interest of this figure. If the culture of fibroblasts has ever been published, delete this figure. If not, the characterization of the cells obtained by this technique of culture should be added.
- Figure 3: Why the x-axis is in log?
- L158 “seen(Fig. 4)..” Add onse space before “(“ and delete one point
Discussion
- “In the present study, we designed and optimized an in vitro model to simulate the mouth-washing process using an antiviral solution.” How have you proved that your model corresponds to the mouth-washing process?
- You hypothesized that B-OT could have an antiviral effect on human fibroblasts infected by SARS-CoV-2. Some questions should be answered to support this hypothesis which is very different from your study. First of all, are there study demonstrated that human fibroblasts of gingival tissue are infected by SARS-CoV-2? If yes, please add references. Moreover, do studies proving the evidence that your concentration of B-OT/ number of cells in culture are relevant of the in vivo situation exist?... Your discussion should be revised to be more based on facts.
L256-258 “In our study, we designed and optimised an in vitro model simulating the mouth rinsing process to explore a possible application of B-OT in the aspiration tract and to investigate the tolerance of primary human gingival fibroblasts to B-OT.” You cultured human gingival fibroblast and not human fibroblasts of the aspiration tract. Do you have evidence of the similarity of
Author Response
Thank you very much and admire your dedication and appreciate your help.
The conclusions of your article are not based on the results of this article. One major remark concerns the tittle which doesn’t reflect the study because you did neither analyzed cells infected with SARS-CoV-2 nor the use of mouthwash. Moreover, the number of samples is very low because you used only one sample for the results of the figure 4.
Reply: This is a very good question you have asked. Before we started this study, one of our studies showed that B-OT has been used in a compassionate use setting to treat a hospitalised COVID-19 patient with confirmed SARS-CoV-2-related pneumonia in both lungs and oxygen deprivation. Prior to B-OT treatment, severe damage to the lungs with marked infiltrates of viral pneumonia was observed. Oral administration of a daily dose of 6mg for 7 days completely suppressed SARS-COV-2 replication (PCR negative), showed a marked decrease in the previously pronounced infiltrates, and resulted in rapid clinical recovery of the patient within 7 days (manuscript in preparation). (https://www.ots.at/presseaussendung/OTS_20211210_OTS0013/neuer-therapieansatz-gegen-sars-cov-2-erfolgreich-getestet)In a patient with 70kg body weight, 6mg B-OT corresponds to a serum peak concentration of 500pM.
Because this drug is effective against the SARS-CoV-2, we are conducting the current experiment to ensure that this drug does not harm normal human fibroblasts; in other words, the concentration of the drug that proves effective against the SARS-CoV-2 has no toxic effect on human cells.
Regarding the experimental sample, we tested seven cases with roughly comparable results. To exclude differences between gingival fibroblasts, our statistics were compared among different groups in the same sample.
Tittle
- The title should be revised because “Benfo-Oxythiamine as potential mouth-wash and nasal spray 2 against SARS-CoV-2” is a hypothesis and not results of this study
Maybe one tittle such as “Tolerance of human fibroblasts to Benfo-Oxythiamineas in an in vitro mouth-washing model”
Reply: Although no data on SARS-CoV-2 virus inhibition was generated in the submitted manuscript, it has been recently published in the MDPI journal Metabolites, as listed above, that B-OT inhibits the SARS-CoV-2 virus. Therefore, I would add slightly to your suggested title so that the link between these experiments and the data recently published in Metabolites is more easily recognisable to the reader and also more indicative of this Metabolites article.
Suggestion: Tolerance of human fibroblasts to the anti-SARS-CoV-2 virus compound Benfo-Oxythiamine as in an in vitro mouth-washing model
Abstract
- L14 Give the meaning of « B-OT” reply: B-OT is abbreviation of Benfo-Oxythiamine. When it first appears in the text, we explain it. Please see line 42.
- L16 “to assess tolerance” Please clarify the tolerance of what? Reply: We changed the original sentence to “We conceived and optimized an in vitro model simulating the mouth-washing process to assess tolerance to B-OT on primary human gingival fibroblasts.”
- L40 “cells [1,2].A human” Add one space after the point. Reply: done as suggestion.
- L54-90 This section should be summarized to be more clear. Reply: done as suggestion
- The section clinical relevance that is not a mandatory of this journal should be deleted because it is purely hypothetic. Reply: done as suggestion
Material and methods
- Add subheading. Reply: done as suggestion.
- The technique of culture of fibroblast has ever been published. Please add references. Reply: done as suggestion.
Results
- Figure 1: I am not sure of the interest of this figure. If the culture of fibroblasts has ever been published, delete this figure. If not, the characterization of the cells obtained by this technique of culture should be added. Reply: Figure 1 is intended to demonstrate the process of obtaining gingival fibroblasts so that it can better guide doctors and scientist without dental expertise to better obtain gingival fibroblasts. After all, primary cells for SARS-CoV-2 drug testing, which is a very good choice. If you think this is not very meaningful, we can remove this set of figures.
- Figure 3: Why the x-axis is in log? Reply: Because in our experiments, the drug concentration units are mM,The four experimental groups, if converted to μM, would become 10 100 1000 10000. So these four experimental groups take log, which can be converted into numbers 1 2 3 4, so that the arrangement in x-axis is more concise.
- L158 “seen(Fig. 4)..” Add onse space before “(“ and delete one point Reply: done as suggestion.
Discussion
- “In the present study, we designed and optimized an in vitro model to simulate the mouth-washing process using an antiviral solution.” How have you proved that your model corresponds to the mouth-washing process?
Reply: When we rinse one's mouth, first hold the mouthwash in mouth. In our experiments the drugs were first added to the culture plates.
Then, when we gargle, we need to move the muscles in the mouth to make the mouthwash vibrate in the mouth. In our experiment, the plates were shaken for 1 min to make the drug vibrate in plate.
We have added the following sentence to the methods section to make this process more understandable to the reader. “After the drug was added to the culture plates, it was shaken for one minute.”
.
- You hypothesized that B-OT could have an antiviral effect on human fibroblasts infected by SARS-CoV-2. Some questions should be answered to support this hypothesis which is very different from your study. First of all, are there study demonstrated that human fibroblasts of gingival tissue are infected by SARS-CoV-2? If yes, please add references. Moreover, do studies proving the evidence that your concentration of B-OT/ number of cells in culture are relevant of the in vivo situation exist?... Your discussion should be revised to be more based on facts.
Reply: SARS-CoV-2 infects epithelial cells of the upper respiratory tract (URT; including the nasal, oral and throat) and the lungs (bronchi and lung alveoli)[1]. That's why all the tests have to oro/nasopharyngeal swab for SARS-COV-2. It is intuitively obvious that the oral cavity, along with the nasal cavity, being proximal to the airway, would be the most direct route for microbes to enter the airway[2]. We will add the reference to the appropriate part.
As decribed in the dicussion, a recent compassionate clinical use of B-OT for a hospitalized COVID-19 patient demonstrated high potency of B-OT with an efficacy dose of 6mg daily, corresponding to a peak serum concentration of approximately 500pM. This effective dose range of BOT is 106–fold below the 1000µM which did not show any toxic effect on cultured primary gingival cells in the present study. Even when B-OT was kept in medium permanently during the 3-day treatment period, the cells were completely toler-ant to 10µM and only slightly affected by 100µM.
- L256-258 “In our study, we designed and optimised an in vitro model simulating the mouth rinsing process to explore a possible application of B-OT in the aspiration tract and to investigate the tolerance of primary human gingival fibroblasts to B-OT.” You cultured human gingival fibroblast and not human fibroblasts of the aspiration tract. Do you have evidence of the similarity of
Reply: This is an great interesting bio-logical question. Upper respiratory tract is normally considered to includethe nasal cavity, the oral cavity, the oro-and nasopharynx, and the larynx[3]. The oral and nasal cavities are covered by the same mucosal epithelium that starts at the beginning of the aero-digestive tract[4,5]. Therefore these cells should be interpreted as same cells.
In addition, the upper respiratory tract, starting with the nasal and oral cavities, is a major route for entry of pathogens into the body[6].
In contrast, a significant number of cells are easy to obtain because we are Oral and Maxillofacial Surgeon.
- Lipsitch, M.; Grad, Y.H.; Sette, A.; Crotty, S. Cross-reactive memory T cells and herd immunity to SARS-CoV-2. Nat Rev Immunol 2020, 20, 709-713, doi:10.1038/s41577-020-00460-4.
- Mammen, M.J.; Scannapieco, F.A.; Sethi, S. Oral-lung microbiome interactions in lung diseases. Periodontol 2000 2020, 83, 234-241, doi:10.1111/prd.12301.
- George, S.C.; Hlastala, M.P. Airway gas exchange and exhaled biomarkers. Compr Physiol 2011, 1, 1837-1859, doi:10.1002/cphy.c090013.
- Kiyono, H.; Yuki, Y.; Nakahashi-Ouchida, R.; Fujihashi, K. Mucosal vaccines: wisdom from now and then. Int Immunol 2021, 33, 767-774, doi:10.1093/intimm/dxab056.
- Miao, M.; Peng, M.; Xing, Z.; Liu, D. Effect of Shuangjinlian mixture on oral ulcer model in rat. Saudi J Biol Sci 2019, 26, 790-794, doi:10.1016/j.sjbs.2019.02.005.
- Saferali, A.; Tang, A.C.; Strug, L.J.; Quon, B.S.; Zlosnik, J.; Sandford, A.J.; Turvey, S.E. Immunomodulatory function of the cystic fibrosis modifier gene BPIFA1. PLoS One 2020, 15, e0227067, doi:10.1371/journal.pone.0227067.

Round 2
Reviewer 1 Report
My query:
Also, your aim was only to check the tolerance of B-OT. It would be misleading if in your response you fail to show the claimed evidence (abstract; line 24).
Your Reply: This is a very good question you have asked. Before we started this study, one of our studies showed that B-OT has been used in a compassionate use setting to treat a hospitalised COVID-19 patient with confirmed SARS-CoV-2-related pneumonia in both lungs and oxygen deprivation. Prior to B-OT treatment, severe damage to the lungs with marked infiltrates of viral pneumonia was observed. Oral administration of a daily dose of 6mg for 7 days completely suppressed SARS-COV-2 replication (PCR negative), showed a marked decrease in the previously pronounced infiltrates, and resulted in rapid clinical recovery of the patient within 7 days (manuscript in preparation).
Upon a careful read, you will know that my query above was on your current investigation. However, your reply was based on your earlier investigation (as highlighted above). Your investigation may not provide a relevant conclusion if you do not address your aim i.e. to check the tolerance of B-OT. Reiterating; it would be misleading if in your response you fail to show the claimed evidence in the current investigation (abstract; line 24).
Reviewer 2 Report
This in vitro research, which appears to be in contradiction with the announced clinical objectives, has no concrete link with SARS-2.
Reviewer 3 Report
Dear author,
Thank you for considering some of my comments
Please delete SRAS-coV-2 from your title because as previously explained you didn't studied cells infected by SARS-CoV-2 and you can't do marketing!